# Acculturation and Disordered Eating among Asian American College Students: The Role of Objectification through a Sociocultural Lens

**DOI:** 10.3390/ijerph192113967

**Published:** 2022-10-27

**Authors:** Zhiqing Zhou, Jeffrey Liew, Wen Luo

**Affiliations:** Department of Educational Psychology, Texas A&M University, College Station, TX 77843, USA

**Keywords:** disordered eating, body image, body shame, self-objectification, acculturation, Asian American

## Abstract

Disordered eating is a public health problem because it’s highly prevalent, dangerous, and costly. More research about its risk factors and mechanisms is needed to address this problem and prevent disordered eating among high-risk populations, particularly understudied ethnic minorities. The present study contributes to the limited existing research on acculturation and disordered eating among Asian American college students who represent an understudied and high-risk group. The sample consisted of 245 Asian American (primarily East and Southeast Asian American) college students who provided data on their acculturation status, internalization of thin and muscular body ideals, body surveillance, body shame, and disordered eating. Results show that after controlling for gender, both cultures are positively associated with internalization of the muscular body ideal, but only the Asian culture of origin is associated with disordered eating. Additionally, path analysis results show that Asian culture of origin has a significant total effect on disordered eating as well as a significant indirect effect on disordered eating, mediated by thin body ideal internalization. While American culture does not have a significant contribution to body ideal internalization or disordered eating, it interacts with Asian culture of origin and put participants with high levels of both cultures at a greater risk for muscular body ideal internalization. Findings highlight the importance of cultural context in the understanding of body experiences and disordered eating among Asian American college students and have implications for the prevention and intervention of these problems in this high-risk population.

## 1. Introduction

Disordered eating is a serious public health problem. It is estimated that over 3.3 million lives are lost to eating disorders every year worldwide [1]. In the United States, lifetime prevalence of eating disorders is about 6.0% in women and 4.07% in men [2], with the prevalence peaking among adolescents and young adults [3]. Asian American young adults are at an elevated risk for symptoms of disordered eating and body image issues compared with Whites and non-Asian people of color [4]. Understanding Asian Americans’ body experiences and eating pathologies is critical to address this health disparity. Furthermore, the majority of Asian Americans were born outside of the U.S. [5] and are actively undergoing acculturation processes. Thus, acculturation is an important factor in the study of disordered eating for Asian Americans. Previous research regarding acculturation and disordered eating has yielded inconsistent results and the mechanism remains unclear. The present study aims to clarify the role of acculturation in disordered eating and contributes to the existing literature by examining the role of objectification as a mediator through which acculturation influences disordered eating among Asian American college students.

### 1.1. Acculturation and Disordered Eating

Acculturation refers to the adaptation process when individuals from one culture are introduced to a new cultural context [6]. It entails two major decisions: to what extent the original cultural identity and traits are maintained and to what extent individuals are involved in the new cultural context. The decisions can lead to four acculturation strategies: assimilation (adopts the new culture and discards the original culture), separation (rejects the new culture and maintains the original culture), integration (adopts the new culture and maintains the original culture), and marginalization (rejects both cultures) [6]. Recent studies have concluded that the strategy of integration is associated with the most positive outcomes, whereas marginalization is associated with the most negative outcomes [7,8].

Despite this well-established relationship between acculturation and developmental outcomes, the relationship between acculturation and disordered eating among Asians and Asian Americans is still inconclusive. Researchers have found positive relations [9,10], negative relations [11], and no relations [12,13] between these two variables. Given these inconsistent results, it could be postulated that there are potential mediators or moderators between acculturation and disordered eating.

### 1.2. Objectification as a Mediator

#### 1.2.1. The Objectification Theory

The objectification theory [14] originally aimed to understand women’s experiences as they live in a society where their bodies are sexually objectified. Over time, this sexual objectification socializes women to internalize certain body ideals and objectify themselves, which is manifested as body surveillance. Body surveillance can then lead to body shame and eating disorders [14]. While the original theory focused on middle-class White women, an amended model that took factors of diversity into account was then proposed and tested in adolescent boys [15] and girls [16], men [17] and women [18], people who identify as heterosexual, gay/lesbian, and bisexual [19,20,21], as well as Black and Hispanic women [22], among other diverse populations. In addition to sexual objectification, other socialization experiences were included as causes of self-objectification as well, such as cultural identity conflict/marginalization and experience of racism [23]. These socialization experiences are particularly pertinent to people who are going through the acculturation process. Exploring and defining identity is a critical task of acculturation [24]. Previous research has also shown that acculturation significantly predicts racism and related stress among Asian Americans [25].

Limited research to date has examined the applicability of objectification theory to Asian Americans. A recent study employed the racially extended model to explain the relationship between experiences of racism and disordered eating among Asian American women [26], which supported the applicability of the theory to Asian Americans. The present study aims to advance the literature by examining whether other pathways of the racially extended model, particularly regarding acculturation and cultural identity, may be relevant to Asian American college students.

#### 1.2.2. Objectification of Asian Americans

The body is a social construct that exists in cultural and social contexts [14]. For Asian Americans, acculturation provides a sociocultural context that impacts the process of objectification and body experiences. Individuals who are undergoing acculturation are often exposed to messages in the media of the mainstream culture. Researchers found that media exposure had a higher influence than family or peers on Asian American women’s body dissatisfaction and attributed it to the underrepresentation of Asian Americans in American media [27]. This underrepresentation indirectly communicates to Asian American women that “[their] type is not reflected in what’s beautiful, what’s popular, or even prevalent” [28]. In the meantime, Asian American women are hyper-sexualized and objectified. On social media, many Asian American women shared their experiences of being perceived as “exotic” and called “China doll” and “Oriental”, regardless of their Asian ethnicities [29]. Exposure to media in the mainstream American culture can adversely affect Asian American men as well. Asians are often represented and perceived as a feminine race, and therefore, Asian American men are even more underrepresented than Asian American women in the media [30]. A common and stereotypical perception of Asian American men is that they are weak and not sexually attractive [29]. Both Asian American women and men experience objectification as they adopt the mainstream American culture, and thus may develop body image issues and disordered eating.

In addition to the influence of the mainstream American culture, Asian Americans experience objectification from their Asian culture of origin, too. Both American and Asian cultures have strict standards for the beauty of women. Asian cultures also have a thin body ideal as part of women’s gender role [31]. While the American body ideal for women has shifted from just being thin to being both thin and muscular [32] and focusing on physical fitness and muscle definition, the Asian ideal still emphasizes skinniness, fragility, and paleness [28]. As for attractiveness of men, Asian standards focus on being tall and of the right amount of muscularity (i.e., “somewhat” muscular, but not “too big”) [33]. It appears that maintaining their Asian cultures can possibly put Asian Americans at risk for body ideal internalization and body image issues as well.

### 1.3. The Present Study

The present study aims to examine the relation between acculturation and disordered eating among Asian American college students, mediated by objectification (i.e., internalization of body ideals, body surveillance, and body shame). Guided by the amended objectification theory, the hypothesized model (Figure 1) was tested. Acculturation is operationalized based on Berry’s theory [6], measured by both the maintenance of Asian culture of origin and the adoption of mainstream American culture. The main effects and the interaction effect between these two cultures were all tested. In addition, both the internalization of thin body ideals and muscular body ideals were included in the model, since recent research has shown that these two body ideals are relevant to both men and women [28,32,33].

## 2. Methods

### 2.1. Procedures and Participants

Participants were recruited online through listserv emails to Asian student organizations and the utilization of a snowball recruitment strategy. To be eligible to participate in the study, participants needed to: (1) self-identify as of Asian descent, (2) born in the U.S. or immigrated to the U.S. before the age of 12 years, (3) enrolled in a college in the U.S. at the time of participation, (4) between 18 and 24 years of age, and (5) able to read English. While recruitment emails were sent nationwide, most participants were attending college in Texas, California, and the New England area. Participants were recruited from approximately 36 colleges and universities.

The sample included 245 Asian American college students (mean age = 20.36, *SD* = 1.58). All participants identified as cisgender. About two-thirds of the participants identified as women (162 women and 83 men). No other gender identification was reported. The majority of participants (*n* = 229, 93.47%) identified as heterosexual, while 13 participants identified as gay/lesbian, nine as bisexual, and one as other. Most participants were born in the U.S. (*n* = 169, 68.98%). On average, participants had spent 18.24 years in the U.S. (*SD* = 3.65). Most participants were East or Southeast Asian. About one-third of the participants identified as of Chinese/Taiwanese descent (*n* = 93, 37.96%), followed by Vietnamese (*n* = 66, 26.94%), Korean (*n* = 60, 24.49%), and Filipino (*n* = 14, 5.71%). The rest of the participants (4.9%) identified as Japanese, South Asian, Indian, Nepalese, etc.

### 2.2. Measures

#### 2.2.1. Asian American Multidimensional Acculturation Scale (AAMAS)

The AAMAS [34] has three subscales that measure participants’ acculturation to European American culture (AAMAS-EA), Asian culture of origin (AAMAS-CO), and pan-ethnic Asian American culture (AAMAS-AA). Each of the subscales has four factors and 15 items, including 6 items measuring Cultural Identity (e.g., “How much do you identify with the White mainstream groups/your own Asian culture of origin/other Asian Groups in America?”), 4 items measuring Language (e.g., “How well do you speak the language of the White mainstream groups/your own Asian culture of origin/other Asian Groups in America?”), 3 items measuring Cultural Knowledge (e.g., “How knowledgeable are you about the culture and traditions of the White mainstream groups/your own Asian culture of origin/other Asian Groups in America?”), and 2 items measuring Food Consumption (e.g., “How often do you eat the food of the White mainstream groups/your own Asian culture of origin/other Asian Groups in America?”). Participants rated each item on a 6-point Likert scale, ranging from *not very much* to *very much*, with higher scores representing a higher level of acculturation. Researchers completed three studies to determine the psychometric properties of AAMAS. The coefficient alpha for AAMAS-EA ranged from 0.76 to 0.81, and for AAMAS-CO, between 0.87 and 0.91 [34]. In the present sample, coefficient alpha of AAMAS-EA is 0.82, and 0.88 for AAMAS-CO.

#### 2.2.2. Sociocultural Attitudes towards Appearance Questionnaire-4 (SATAQ-4)

Internalization of thin and muscular body ideals was measured by the SATAQ-4 [35]. Each of the subscales has 5 items. Sample items include “I want my body to look like it has little fat”, and “it’s important for me to look athletic”. Participants responded on a 5-point Likert scale, ranging from *definitely disagree* to *definitely agree*. Higher scores indicate higher internalization of body ideals. Cronbach’s alphas are 0.82–0.92 among U.S. women, 0.89–0.91 among non-U.S. women, and 0.75–0.90 among men [35]. In the present sample, the coefficient alpha for the thin body ideal is 0.70 and for the muscular body ideal is 0.88.

#### 2.2.3. Objectified Body Consciousness Scale (OBCS)

Body surveillance and body shame were measured by the OBCS [36]. There are eight items in each of the subscales. A sample item from the Surveillance subscale is “during the day, I think about how I look many times”. A sample item from the Body Shame subscale is “I feel ashamed of myself when I haven’t made the effort to look my best”. Participants reported their responses on a 7-point Likert scale, ranging from *strongly disagree* to *strongly agree*, with higher scores representing higher respective features of objectified body consciousness. Cronbach’s alphas are within the range of 0.77–0.89 for the Surveillance subscale and 0.75–0.84 for the Body Shame subscale among men and women from diverse cultural backgrounds [36,37]. In the present study, coefficient alphas are 0.86 for Surveillance and 0.83 for Body Shame.

#### 2.2.4. Eating Disorder Examination—Questionnaire, Version 6.0 (EDE-Q 6.0)

The EDE-Q 6.0 [38,39] is widely used to obtain descriptive information about eating disorder symptoms in the past 28 days. It has 22 items, including items such as “how uncomfortable have you felt seeing your body” and “have you had a definite fear of losing control over eating”. Participants responded on a 7-point Likert scale (0–6), with higher ratings indicating more severe eating disorder symptoms. At the end of the scale, there were additional questions asking participants to give their best estimate of their weight at present and height. Self-reported height and weight were used to calculate BMI. In a sample of Asian American and Latina college students, Cronbach’s alpha is 0.95 for the EDE-Q full scale [40]. In the present sample, coefficient alpha of the full scale is 0.94.

### 2.3. Data Analyses

Data analyses were performed in R [41]. Descriptive statistics and correlational analyses were conducted using the *plyr* [42], *psych* [43], and *MVN* [44] packages. Path analyses were conducted using the *lavaan* [45] package. Descriptive analyses were conducted first, including mean, standard deviation, and normality (skewness and excess Kurtosis). Gender differences on all major variables were tested by MANOVA, followed up with univariate ANOVAs to check which individual variables differ between gender groups. Then, correlational analyses were conducted to test relations between major variables. For path analyses, the maximum likelihood estimation with robust standard errors was used to estimate all path coefficients. To test direct and indirect effects, bootstrap confidence interval method was employed. There were no missing data in the dataset.

## 3. Results

### 3.1. Descriptive Analyses

All major variables’ mean, *SD*, skewness, and excess Kurtosis in the total sample are shown in Table 1. BMI is approaching the thresholds of moderate nonnormality (BMI Skewness = 1.61, approaching 2.0, and excess Kurtosis = 4.48, approaching 7.0), which could potentially become problematic [46]. Therefore, a log transformation was performed.

Descriptive statistics were also examined by gender. A MANOVA test was performed to test for gender differences on major variables. Results indicate that there are significant gender differences between the major variables (Pillai’s Trace statistic = 0.36, *F* (8, 236) = 16.93, *p* < 0.001). Subsequent univariate ANOVA tests were conducted. Using Bonferroni’s correction, the test-wise alpha level was adjusted to 0.00625. Results show significant gender differences on the thin body ideal (*F* (1, 243) = 10.96, *p* = 0.001, *d* = 0.45), muscular body ideal (*F* (1, 243) = 51.10, *p* < 0.001, *d* = 0.96), EDE-Q (*F* (1, 243) = 10.64, *p* = 0.001, *d* = 0.42), and log-transformed BMI (*F* (1, 243) = 10.73, *p* = 0.001, *d* = 0.42). Overall, women endorse higher levels of internalization of the thin body ideal and disordered eating, while men endorse higher levels of internalization of the muscular body ideal and BMI.

### 3.2. Correlational Analyses

Bivariate and partial correlations (controlling for participant gender) are presented in Table 2. After controlling for gender, maintenance of Asian culture of origin is positively associated with internalization of the muscular body ideal (*r* = 0.13, *p* = 0.04) and disordered eating (*r* = 0.13, *p* = 0.04). Adoption of American culture is also positively associated with the muscular body ideal (*r* = 0.13, *p* = 0.04). Thin and muscular body ideals were positively associated with each other (*r* = 0.38, *p* < 0.001). While both body ideal internalization are positively associated with body surveillance (thin body ideal: *r* = 0.38, *p* < 0.001, and muscular body ideal: *r* = 0.14, *p* = 0.03), body shame (*r* = 0.46, *p* < 0.001, and *r* = 0.27, *p* < 0.001), and disordered eating (*r* = 0.60, *p* < 0.001, and *r* = 0.31, *p* < 0.001), only thin body ideal internalization is positively associated with log-transformed BMI (*r* = 0.20, *p* = 0.002). Body surveillance and body shame are positively associated with each other (*r* = 0.40, *p* < 0.001) and with disordered eating (*r* = 0.41, *p* < 0.001, and *r* = 0.69, *p* < 0.001, respectively). Log-transformed BMI is positively associated with body shame (*r* = 0.28, *p* < 0.001) and disordered eating (*r* = 0.46, *p* < 0.001), in addition to thin body ideal internalization mentioned above.

### 3.3. Path Analyses

The hypothesized model (Figure 1) was specified. Log-transformed BMI was included in the model instead of BMI. The original model was not a good fit, and therefore, post hoc modifications were conducted to improve the fitness of the model to the data. After consulting the modification indices, correlated errors were found between the internalization of the thin body ideal and the muscular body ideal. After adding this correlated path to the hypothesized model, it has improved the model fit. Model fit indices suggest that the modified model is a good fit (*χ*^2^(11) = 19.40, *p* = 0.05, RMSEA = 0.06, SRMR = 0.04, CFI = 0.98). Standardized model results are shown in Figure 2. Overall, the model accounts for 65% of the variance in disordered eating.

Total, direct, and indirect effects were examined to test research hypotheses. The total effect of maintenance of Asian culture of origin on disordered eating is statistically significant (*β* = 0.23, *p* = 0.01, CI = [0.06, 0.41]). One significant indirect effect was found, from maintenance of Asian culture of origin to thin body ideal internalization, and then to disordered eating (*β* = 0.06, *p* = 0.04, CI = [0.01, 0.13]). There were two other indirect effects that were not statistically significant, yet noteworthy. The first indirect effect was from Asian culture of origin to thin body ideal, to body shame, and then to disordered eating (*β* = 0.03, *p* = 0.052, CI = [0.004, 0.061]). The second indirect effect was from Asian culture of origin to thin body ideal, to surveillance, and then to body shame and disordered eating (*β* = 0.01, *p* = 0.06, CI = [0.001, 0.024]). It appears that the internalization of the thin body ideal can mediate the relation between the maintenance of Asian culture of origin and disordered eating.

Additionally, the interaction effect between the two cultures has a unique and significant contribution to the internalization of muscular body ideal (*β* = 0.15, *p* = 0.01, CI = [0.04, 0.25]). The main effect of Asian culture of origin is also significant (*β* = 0.12, *p* = 0.04, CI = [0.01, 0.24]), but the main effect of American culture is not significant (*β* = 0.10, *p* = 0.08, CI = [−0.01, 0.21]). These results suggest that Asian culture has an influence on the internalization of the muscular body ideal; participants who maintain more of their Asian culture of origin are more likely to have a stronger internalization of the muscular body ideal. While American culture does not have an influence over the internalization of the muscular body ideal, it can interact with Asian culture of origin, and therefore, participants with high levels of both cultures are at the greatest risk of muscular body ideal internalization.

## 4. Discussion

The primary purpose of the present study is to examine the role of objectification as a mediating mechanism through which acculturation influences disordered eating among Asian American college students. Results confirm the mediating role of objectification and suggest that Asian cultures of origin have more contributions to self-objectification and disordered eating than the mainstream American culture in Asian American emerging adults.

### 4.1. Heritage Cultures and Objectification

Results of the correlation analyses suggest that the maintenance of Asian cultures of origin co-occurs with the internalization of body ideals and disordered eating. Most importantly, results from the path analyses show that holding onto heritage cultures has a significant and unique contribution to thin and muscular body ideal internalization and an indirect contribution to disordered eating. This is consistent with previous research showing that individuals of Asian descent experience significant body dissatisfaction and are at risk for disordered eating [4,47,48].

A possible explanation for these results might be the standards for beauty or physical attractiveness among Asian cultures. As mentioned in the literature review, Asian standards of attractiveness focus on skinniness, fragility, and paleness for women [28] and on being tall and of the right amount of muscularity for men [33]. Thus, Asian American young adults may internalize these body ideals and develop eating disturbance when they maintain their Asian cultures of origin.

Another possible explanation might be that specific Asian values may contribute to the internalization of body ideals and eating disturbance among Asian American young adults. Individuals of Asian descent often value interpersonal relatedness and collectivism [49]. People who subscribe to collectivistic values are interdependent with other members in their social groups (i.e., family, peer group, etc.), construct their identities in relation to their groups, and internalize and conform to the group norms [50]. Since body ideals are a part of cultural norms, it is plausible that Asian American individuals may readily internalize body ideals due to collectivistic values when they maintain their Asian cultures of origin.

### 4.2. American Culture Unrelated to Objectification

Contrary to our expectations, results show that the adoption of American culture is not related to objectification and disordered eating for Asian American young adults. As discussed in the literature review, previous research has shown mixed results regarding the association between American culture and disordered eating [12,13,48], and some studies have found that the adoption of American culture, or Western culture in general, is not associated with disorder eating [13,31]. Therefore, there might be some factors that could potentially moderate this association. A possible example would be messages people are receiving from the media. On the one hand, mainstream American culture imposes body ideals on both women and men through mass media and other societal influences. On the other hand, there have been social movements promoting body positivity and body neutrality on social media in recent years. With the body positivity movement, diverse body attributes are depicted in social media [51]. The representation of diverse body attributes sends the message that all body types need to be respected and accepted. There is also an emerging discussion about body neutrality. Body neutrality further encourages people to avoid placing too much value on their appearance [52]. Moving away from the appearance focus could potentially prevent the objectification of the body. These social movements can serve as a protective factor to support individuals’ resistance against the development of body image problems and disordered eating. Therefore, depending on whether individuals have more exposure to traditional body ideals or recent body image movements, adopting American culture could have differential effects on their body experiences.

### 4.3. The Interaction Effect between Asian and American Cultures

Results show a significant interaction effect between the Asian and American cultures on the internalization of the muscular body ideal but not on the internalization of the thin body ideal. Contrary to our expectations and inconsistent with previous research, integration/high levels of both cultures appear to be associated with an elevated risk of muscular body ideal internalization among Asian American college students.

Previous research associates integration with positive developmental outcomes partly because integration may be related to increased flexibility and adaptability in response to stressors [7,8,53]. For Asian American individuals, however, this benefit of integrating and switching between different cultural norms does not seem to apply to their internalization of the muscular body ideal, as suggested by the results of the present study. Since both Asian and American cultures have different yet similar muscular body ideals, being immersed in the two cultures may reinforce the internalization of the body ideal instead of allowing individuals to integrate and switch between diverse body ideals. On the contrary, even though both cultures have thin body ideals, the recent movement of body positivity and body neutrality may allow Asian Americans to switch between traditional thin body ideal in the Asian cultures of origin and diverse body types portrayed on social media, and thus mitigate the risk of internalizing the thin body ideal.

### 4.4. Limitations

While this is one of the few known studies to examine the role of acculturation status on the process of objectification and disordered eating, the present study findings should be interpreted with several study limitations in mind. The major limitation of this study is the cross-sectional design, which makes it impossible to draw causal conclusions regarding the association between acculturation and disordered eating among Asian American college students. Second, although Asian American ethnic groups share some common values, there are still substantial within-group differences among Asian Americans. The present study captured a broad range of Asian ethnic groups, including Chinese/Taiwanese, Vietnamese, Korean, Filipino, Japanese, Indian, etc. However, it still does not represent all Asian Americans in the U.S. Third, the present study used the overall score of the AAMAS to quantify participants’ acculturation level. Future research may consider looking at the interactions between behaviors and identity associated with the two cultures, as well as the impact of cultural values, when examining the influence of acculturation on disordered eating. Lastly, participant BMI was calculated based on self-reported height and weight, which would be less accurate than objectively measured height and weight.

## 5. Conclusions 

The aim of the present study is to examine the role of acculturation in the development of disordered eating among Asian American young adults, with objectification, which includes internalization of body ideals, body surveillance, and body shame, being the mediating mechanism. Results comfirmed the mediating role of objectification. As for effects of acculturation, maintaining heritage culture has a direct contribution to thin and muscular body ideal internalization and an indirect contribution to disordered eating, while adopting mainstream American culture does not contribute to objectification or disordered eating. An interesting result is regarding the interaction effect between the two cultures. Bicultural Asian American young adults appear to be at a high risk for the internalization of muscular body ideal. These findings point to possible clinical considerations for body image and disordered eating among Asian American emerging adults. 

Therefore, it is crucial for researchers and practitioners to consider acculturation processes such as acculturative stress and explicitly assess sociocultural factors as part of a comprehensive conceptualization of Asian American individuals’ body experiences. For example, there is a need for research on acculturative stress as well as internalized racism regarding being of Asian descent in a society where their natural body type and features are not valued and are considered unattractive. It is also important for future research on bicultural individuals’ experience of objectification and disordered eating to examine their exposure, consumption, and internalization of heritage and American content through mass and social media. Additionally, there is a need for research that uses quantitative and qualitative methods to understand bicultural individuals’ heritage and American body ideals and how they navigate the differences and similarities between them, particularly surrounding collectivism, conformity with social norms, and sources of shame.

## Figures and Tables

**Figure 1 ijerph-19-13967-f001:**
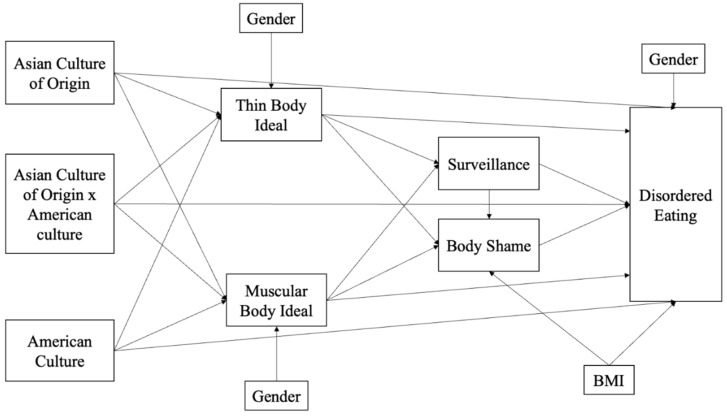
Proposed Path Model of the Relations between Acculturation, Objectification, and Disordered Eating (BMI: Body Mass Index).

**Figure 2 ijerph-19-13967-f002:**
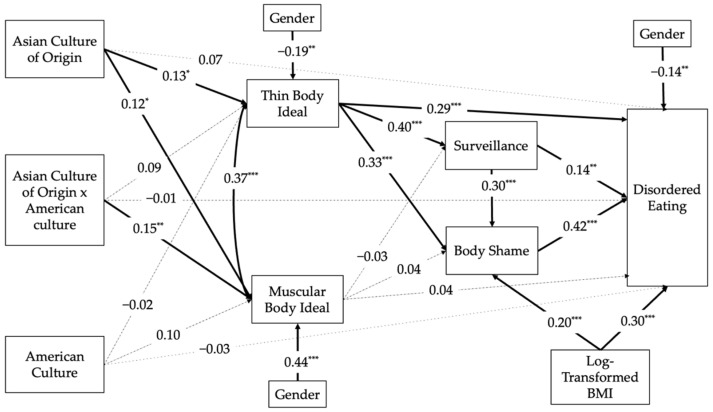
Standardized Model Results of the Relations between Acculturation, Objectification, and Disordered Eating (BMI: Body Mass Index. Solid lines represent statistically significant paths. * *p* < 0.05. ** *p* < 0.01. *** *p* < 0.001).

**Table 1 ijerph-19-13967-t001:** Descriptive Statistics and Gender Differences of Acculturation, Objectification, Disordered Eating, and Body Mass Index.

Variables	Total Sample (*n* = 245)				Women (*n* = 162)		Men (*n* = 83)		*F* (1, 243)	*p*
	Mean	*SD*	Skewness	Excess Kurtosis	Mean	*SD*	Mean	*SD*		
Asian Culture of Origin	4.60	0.79	−0.48	0.24	4.70	0.75	4.41	0.82	7.19	0.008
American Culture	4.46	0.67	−0.22	−0.40	4.48	0.67	4.43	0.68	0.22	0.64
Thin Body Ideal	3.44	0.76	−0.08	−0.13	3.55	0.79	3.22	0.66	10.96	0.001
Muscular Body Ideal	2.89	1.01	0.05	−0.76	2.59	0.93	3.47	0.90	51.10	<0.001
Surveillance	3.91	0.90	−0.16	−0.37	3.99	0.85	3.74	0.98	4.27	0.04
Body Shame	3.08	1.00	0.47	−0.21	3.19	1.03	2.86	0.89	5.92	0.02
Disordered Eating	1.82	1.28	0.72	−0.26	2.00	1.29	1.45	1.17	10.64	0.001
Body Mass Index	22.50	3.75	1.61	4.48	21.96	3.49	23.54	4.04	10.73	0.001

Note: Acculturation scores range from 1 to 6. Internalization scores range from 1 to 5. Surveillance and Body Shame scores range from 1 to 6. Disordered Eating scores range from 0 to 6. Log-transformed BMI was used for the ANOVA analysis instead of BMI.

**Table 2 ijerph-19-13967-t002:** Bivariate and Partial Correlations Between of Acculturation, Objectification, Disordered Eating, and Log-transformed Body Mass Index.

Variables		1	2	3	4	5	6	7	8
1	Asian Culture of Origin	--	0.11	0.12	0.13 *	0.08	0.04	0.13 *	−0.03
2	American Culture	0.11	--	-0.01	0.13 *	0.01	−0.01	−0.01	0.01
3	Thin Body Ideal	0.16 *	0.001	--	0.38 ***	0.38 ***	0.46 ***	0.60 ***	0.20 **
4	Muscular Body Ideal	0.05	0.10	0.26 ***	--	0.14 *	0.27 ***	0.31 ***	0.12
5	Surveillance	0.10	0.01	0.39 ***	0.07	--	0.40 ***	0.41 ***	−0.02
6	Body Shame	0.07	−0.003	0.48 ***	0.18 **	0.41 ***	--	0.69 ***	0.28 ***
7	Disordered Eating	0.16 *	−0.01	0.61 ***	0.19 **	0.43 ***	0.69 ***	--	0.46 ***
8	Log-transformed Body Mass Index	−0.06	0.01	0.14 *	0.19 **	−0.05	0.23 ***	0.40 ***	--

Note: Bivariate correlations are provided below the diagonal. Partial correlations are controlled for participant gender. * *p* < 0.05. ** *p* < 0.01. *** *p* < 0.001.

## Data Availability

The data presented in this study are available on request from the corresponding author.

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
