# Peer review of "Acculturation and Disordered Eating among Asian American College Students: The Role of Objectification through a Sociocultural Lens"

_ijerph, 2022, doi:10.3390/ijerph192113967_

Round 1

Reviewer 1 Report

It was a delight to read, “Acculturation and Disordered Eating Among Asian American College Students: The Role of Objectification through a Sociocultural Lens”. This study was very interesting to read, very well written, and fills an important gap in the literature. However, I think the study could be strengthened by addressing the following points:

Introduction:

·        -- It would be helpful to describe the four acculturation strategies and what they are, especially since you go on to disentangle which are associated with positive and negative outcomes

·      --   I’m getting a little lost in the relationship between objectification and acculturation. It may be helpful to readers to have a more definitive explanation about this relationship at the end of section 1.2.1 before specifically digging into the Asian American literature.

·       --  I would like to see more explanation of the theoretically underpinning for men and genders other than cis-women.

·     --    More explanation of why this is particularly relevant among college students since that’s the population for the present study

·      --   Love the inclusion of a thorough theoretical model! However, there is a lot going on here and it would be helpful to bold or somehow highlight what specific associations you were examining in the present study because it cannot all be done.

Methods:

·       --  It would be helpful to have more information on the listservs that were used. For example, were they general university listservs, specific to Asian American clubs or pages, etc. context will be important for interpreting findings.

·       --  Please report all demographics, not just a single group. For example, were there any other genders reported, besides men and women? What % reported identities other than heterosexual? This has been highlighted as an important area of need for the eating disorder literature.

·      --   In relation to my point above re: the conceptual model – this feels like far too much for a single study considering the sample size is small and it is a cross-sectional study. Importantly, there also needs to be consideration for multiple tests in order to minimize potential for type 1 error given the number of tests being conducted. Methodologic changes for type 1 error are needed, and more justification is needed on why the number of tests are appropriate given the concerns mentioned above.

·       --  The SATAQ-4 was validated based on the EDE-Q, particularly the EDE-Q has sub-scales including shape and weight concern so the theoretical model testing the relationship between body ideals and disordered eating seems highly problematic.

·       --  How is there no missing data in the dataset? Not a single question was missed by any of the participants? How?

Results

·        -- It is hard to see where arrows are starting and ending because there are so many here (e.g., arrow between muscular and thin body ideal). It would also be helpful to have bolded arrows if it’s significant b/c there are so many arrows/numbers.

-   -- nicely presented results.

Discussion

·      --   In re: to the paragraph beginning on line 288, does this mean that Asian Americans have decreased risk compared to those of Asian decent not living in America and therefore have not undergone acculturation?

·    --     For section 4.2, I am confused by the logic and arguments here. Western culture and media consumption among college students has continuously been linked with increased likelihood of disordered eating. Further, the literature on body positivity shows mixed results on whether it helpful, harmful, or both. More clarity is needed here.

Nice job to the authors.

Author Response

Response to Reviewer 1 Comments:

Introduction:

  1. It would be helpful to describe the four acculturation strategies and what they are, especially since you go on to disentangle which are associated with positive and negative outcomes.
    • Thank you for pointing this out! We have added brief descriptions of the four strategies on page 2, lines 51-53: “assimilation (adopts the new culture and discards the original culture), separation (rejects the new culture and maintains the original culture), integration (adopts the new culture and maintains the original culture), and marginalization (rejects both cultures).”

  1. I’m getting a little lost in the relationship between objectification and acculturation. It may be helpful to readers to have a more definitive explanation about this relationship at the end of section 1.2.1 before specifically digging into the Asian American literature.
    • We have accordingly clarified the relation between acculturation and objectification on page 2, lines 74-78: “These socialization experiences are particularly pertinent to people who are going through the acculturation process. Exploring and defining identity is a critical task of acculturation. Previous research has also shown that acculturation significantly predicts racism and related stress among Asian Americans.”

  1. I would like to see more explanation of the theoretically underpinning for men and genders other than cis-women.
    • Thank you for this suggestion! On page 2, lines 70-72, we explained that the amended model of objectification theory took factors of diversity into account, and is an inclusive theory that has been tested in populations other than cis-women.

  1. More explanation of why this is particularly relevant among college students since that’s the population for the present study
    • We have added a brief explanation on page 1, line 33 that the prevalence of disordered eating peaks in adolescents and young adults.

  1. Love the inclusion of a thorough theoretical model! However, there is a lot going on here and it would be helpful to bold or somehow highlight what specific associations you were examining in the present study because it cannot all be done.
    • Thank you for this feedback! We indeed tested all of the paths shown in Figure 1. We examined the mediating effect of objectification (i.e. internalization of two body ideals, surveillance, and body shame) between acculturation (i.e. Asian culture, American culture, and their interaction) and disordered eating.

Methods:

  1. It would be helpful to have more information on the listservs that were used. For example, were they general university listservs, specific to Asian American clubs or pages, etc. context will be important for interpreting findings.
    • Agreed! We have edited page 3 line 127 and added that the emails were sent to listservs of various Asian student organizations.

  1. Please report all demographics, not just a single group. For example, were there any other genders reported, besides men and women? What % reported identities other than heterosexual? This has been highlighted as an important area of need for the eating disorder literature.
    • We appreciate this feedback. On page 4 lines 136-139, we added that all participants were cisgender men/women and no other genders were reported. Among participants who did not identify as heterosexual, 13 were gay/lesbian, nine were bisexual, and one was other.

  1. In relation to my point above re: the conceptual model – this feels like far too much for a single study considering the sample size is small and it is a cross-sectional study. Importantly, there also needs to be consideration for multiple tests in order to minimize potential for type 1 error given the number of tests being conducted. Methodologic changes for type 1 error are needed, and more justification is needed on why the number of tests are appropriate given the concerns mentioned above.
    • Thank you for bringing up these concerns! For the first concern re: sample size, the sample size of 245 is not a small sample size since the model is a path model without latent variables. In fact, a simulation study shows that the sample size of 100 is sufficient to obtain accurate standard error estimates in a mediation model with four mediators (MacKinnon, 2000).
    • We acknowledge that the cross-sectional design is a major limitation of our study. This is discussed on page 9, lines 367-370: “The major limitation of this study is the cross-sectional design, which makes it impossible to draw causal conclusions regarding the association between acculturation and disordered eating among Asian American college students.”
    • As for the third concerns re: multiple tests, there is a lack of consensus over when alpha levels should be adjusted. According to Rubin (2021), there are three types of multiple testing: disjunction testing, conjunction testing, and individual testing. All of our null hypotheses surrounding the path model are individual null hypotheses rather than joint null hypotheses, so our tests should be considered as individual testing which does not require the adjustment of alpha. This is also consistent with the common practice in path analyses in which the alpha level for testing individual path coefficients is not adjusted.
    • While it is not necessary to adjust the alpha level for the path analyses, we think Reviewer 1 has brought up a valid concern and thus we adjusted the alpha level for the multiple ANOVA analyses for gender differences. We used the Bonferroni’s correction to set the test-wise alpha level at .05/8 = .00625. There are no statistically significant differences between men and women on Asian Culture of Origin, Surveillance, and Body Shame after the adjustment. These edits are reflected on page 5, lines 221-226. We thank Reviewer 1 again for bringing this to our attention.
    • References: 
      • MacKinnon, D. P. (2000). Contrasts in multiple mediator models. In J. Rose, L. Chassin, C. C. Presson, & S. J. Sherman (Eds.), Multivariate applications in substance use research: New methods for new questions (pp. 141-160). Mahwah, NJ: Erlbaum
      • Rubin, M. (2021). When to adjust alpha during multiple testing: A consideration of disjunction, conjunction, and individual testing. Synthese. https://doi.org/10.1007/s11229-021-03276-4

  1. The SATAQ-4 was validated based on the EDE-Q, particularly the EDE-Q has sub-scales including shape and weight concern so the theoretical model testing the relationship between body ideals and disordered eating seems highly problematic.
    • Theoretically, the SATAQ-4 internalization subscales measure respondents’ desire to obtain a certain figure. The EDE-Q shape and weight concern subscales measure respondents’ concerns about their shape and weight. These constructs, while related, are distinct from each other. In fact, many previous studies have used SATAQ-4 and EDE-Q to examine the relation between body ideals and disordered eating. Here are some examples:
      • Akoury, L. M., Warren, C. S., & Culbert, K. M. (2019). Disordered eating in Asian American women: Sociocultural and culture-specific predictors. Frontiers in psychology10, 1950.
      • Jankauskiene, R., Baceviciene, M., & Trinkuniene, L. (2020). Examining body appreciation and disordered eating in adolescents of different sports practice: cross-sectional study. International journal of environmental research and public health17(11), 4044.
      • Racine, S. E., & Martin, S. J. (2016). Exploring divergent trajectories: Disorder-specific moderators of the association between negative urgency and dysregulated eating. Appetite103, 45-53.
      • Racine, S. E., & Martin, S. J. (2017). Integrating eating disorder-specific risk factors into the acquired preparedness model of dysregulated eating: A moderated mediation analysis. Eating Behaviors24, 54-60.
      • Rodgers, R. F., Slater, A., Gordon, C. S., McLean, S. A., Jarman, H. K., & Paxton, S. J. (2020). A biopsychosocial model of social media use and body image concerns, disordered eating, and muscle-building behaviors among adolescent girls and boys. Journal of youth and adolescence49(2), 399-409.
      • Vartanian, L. R., Hayward, L. E., Smyth, J. M., Paxton, S. J., & Touyz, S. W. (2018). Risk and resiliency factors related to body dissatisfaction and disordered eating: The identity disruption model. International Journal of Eating Disorders51(4), 322-330.

  1. How is there no missing data in the dataset? Not a single question was missed by any of the participants? How?
    • Participants were instructed to answer all the questions. However, acknowledging that some of the items might be sensitive and triggering, it was also explained to them that participation was voluntary and that they could leave the survey at any time. Data were not saved if they left the survey before submitting it. When they submitted the survey, all the missed items were highlighted for participants to review and complete.

Results

  1. It is hard to see where arrows are starting and ending because there are so many here (e.g., arrow between muscular and thin body ideal). It would also be helpful to have bolded arrows if it’s significant b/c there are so many arrows/numbers.
    • Thank you for your suggestion! We have updated the figure with bolded arrows for significant paths. We hope the figure is now easier to read.

  1. nicely presented results.
    • Thank you!

Discussion

  1. In re: to the paragraph beginning on line 288, does this mean that Asian Americans have decreased risk compared to those of Asian decent not living in America and therefore have not undergone acculturation?
    • Our results show that Asian Culture of Origin has a significant positive effect on Thin Body Ideal and Muscular Body Ideal internalization as well as disordered eating, which suggest that among Asian American young adults, those who are lower on their Asian Culture of Origin have decreased risk compared to those who are higher on their Asian Culture of Origin. All other things being equal, it is a reasonable implication that Asians living in Asia would have high endorsement of their respective Asian culture and thus at increased risk for disordered eating.
    • However, Asians living in Asia and Asian Americans have very different cultural experiences, such as the experience of racism, acculturation gap (or lack thereof) within the family, etc., and therefore, conclusions drawn from an Asian American sample may not be directly applicable to Asians living in Asia.

  1. For section 4.2, I am confused by the logic and arguments here. Western culture and media consumption among college students has continuously been linked with increased likelihood of disordered eating. Further, the literature on body positivity shows mixed results on whether it helpful, harmful, or both. More clarity is needed here.
    • Thank you for pointing this out. These results were contrary to our expectations as well! We have included some studies that also found no relation between Western culture and disordered eating in our discussion on page 8, lines 326-328.
    • We also further elaborated on the possible protective effect of body positivity and body neutrality. We wrote “With the body positivity movement, diverse body attributes are depicted in social media. The representation of diverse body attributes sends the message that all body types need to be respected and accepted. There is also an emerging discussion about body neutrality. Body neutrality further encourages people to avoid placing too much value on their appearance. Moving away from the appearance focus could potentially prevent the objectification of the body.”

  1. Nice job to the authors.
    • Thank you and we appreciate your feedback.

Reviewer 2 Report

The authors examine the effect of acculturation on disordered eating among Asian American college students. They showed that Asian and American cultures are positively associated with the internalization of the muscular body ideal, but only the Asian culture of origin is associated with disordered eating. Path analysis shows that, compared to American culture, Asian culture contributes to disordered eating. This study emphasizes the importance of cultural context in the understanding of body experiences and disordered eating among Asian Americans. However, there are a few concerns to be addressed.

In the methods part, they set inclusion criteria as "born in the U.S. or immigrated to the U.S. 124 before the age of 12 years." Could you explain why authors set the limit of age at 12 years?

Could you explain the details of how to measure body mass index? If you calculate BMIs with self-reported high and weight, it would be better to discuss this matter because self-reported hight and weight are not reliable. 

Could you add a table to show the results of univariate ANOVA tests, which were done to test the effect of gender on variables?

Since there is a significant effect of gender on various major variables, could you do the correlation analysis in women and men separately? I know authors controlled for participant gender when they perform these analyses, but still, it would be worth doing these correlation analyses in each gender group separately.

I'm also curious about the results of path analysis in women and men separately.

Could you explain why the authors used log-transformed BMI, but not BMI, for the correlation analysis or path analysis?

Author Response

Response to Reviewer 2’s Comments:

  1. In the methods part, they set inclusion criteria as "born in the U.S. or immigrated to the U.S. 124 before the age of 12 years." Could you explain why authors set the limit of age at 12 years?
    • This is a great question! Previous research about age of immigration shows that immigrants arrived before the age of 12 has similar lifetime risk for mood, anxiety, substance abuse, and other psychiatric disorders as their US-born peers (Takeuchi et al., 2007). In contrast, immigrants arrived after the age of 12 have an advantage regarding mental health over their US-born peers (Breslau, Borges, Hagar, Tancredi, & Gilman, 2009). Therefore, 12 years of age is a reasonable cutoff for our study.
    • References:
      • Breslau, J., Borges, G., Hagar, Y., Tancredi, D., & Gilman, S. (2009). Immigration to the USA and risk for mood and anxiety disorders: variation by origin and age at immigration. Psychological medicine39(7), 1117-1127.
      • Takeuchi, D. T., Zane, N., Hong, S., Chae, D. H., Gong, F., Gee, G. C., ... & Alegría, M. (2007). Immigration-related factors and mental disorders among Asian Americans. American journal of public health97(1), 84-90.

  1. Could you explain the details of how to measure body mass index? If you calculate BMIs with self-reported high and weight, it would be better to discuss this matter because self-reported hight and weight are not reliable. 
    • Thank you for bringing it up! We have edited section 2.2.4 and added that self-reported height and weight were collected as part of EDE-Q. We acknowledge that self-reported height and weight might be unreliable and added it as one of the limitations of the study on page 9, lines 378-380: “Lastly, participant BMI was calculated based on self-reported height and weight, which would be less accurate than objectively measured height and weight.”

  1. Could you add a table to show the results of univariate ANOVA tests, which were done to test the effect of gender on variables?
    • Two columns summarizing the results of univariate ANOVA tests was added to Table 1.

  1. Since there is a significant effect of gender on various major variables, could you do the correlation analysis in women and men separately? I know authors controlled for participant gender when they perform these analyses, but still, it would be worth doing these correlation analyses in each gender group separately.
    • Thank you for this suggestion! Testing the correlations in each gender group separately is a good alternative. We chose to do the partial correlation analyses instead because our primary research questions focused on the relations between acculturation, objectification, and disordered eating. The influence of gender is not part of the main research questions. In addition, objectification theory has shown to be applicable to both men (e.g., Heath, Tod, Kannis-Dymand, & Lovell, 2016) and women (e.g., Cheng, Tran, Miyake, & Kim, 2017), which suggests that while gender differences of the variables exist, the relations between the variables are comparable across genders.
    • References:
      • Cheng, H. L., Tran, A. G., Miyake, E. R., & Kim, H. Y. (2017). Disordered eating among Asian American college women: A racially expanded model of objectification theory. Journal of Counseling Psychology64(2), 179.
      • Heath, B., Tod, D. A., Kannis-Dymand, L., & Lovell, G. P. (2016). The relationship between objectification theory and muscle dysmorphia characteristics in men. Psychology of Men & Masculinity17(3), 297.

  1. I'm also curious about the results of path analysis in women and men separately.
    • This is a great point! However, there were only 83 men participated in the study. With this sample size, we won’t be able to obtain accurate standard error estimates or meaningfully interpret the results.

  1. Could you explain why the authors used log-transformed BMI, but not BMI, for the correlation analysis or path analysis?
    • As we have explained in section 3.1, BMI was approaching the thresholds of moderate nonnormality (BMI Skewness = 1.61, approaching 2.0, and excess Kurtosis = 4.48, approaching 7.0), which could potentially become problematic. Therefore, a log transformation was performed. After the transformation, the Skewness and excess Kurtosis were acceptable (0.90 and 1.67, respectively).

Round 2

Reviewer 1 Report

Thank you for your thoughtful response to comments. Excellent work on this important study.